# A New Construction of $4^{q}$-QAM Golay Complementary Sequences

**DOI:** 10.3390/s22187092

**Published:** 2022-09-19

**Authors:** Gang Peng, Zhiren Han, Dewen Li

**Affiliations:** 1College of Information and Communication Engineering, Harbin Engineering University, Harbin 150001, China; 2Wuhan Maritime Communication Research Institute, Wuhan 430200, China

**Keywords:** orthogonal frequency division multiplexing (OFDM), Golay complementary sequences (GCSs), quadrature amplitude modulation (QAM), generalized Boolean function (GBF), peak-to-mean envelope power ratio (PMEPR)

## Abstract

Quadrature amplitude modulation (QAM) constellation and Golay complementary sequences (GCSs) are usually applied in orthogonal frequency division multiplexing (OFDM) systems to obtain a higher data rate and a lower peak-to-mean envelope power ratio (PMEPR). In this paper, after a sufficient search of the literature, it was found that increasing the family size is an effective way to improve the data rate, and the family size is mainly determined by the number of offsets in the general structure of QAM GCSs. Under the guidance of this idea, we propose a new construction for $4^{q}$-QAM GCSs through generalized Boolean functions (GBFs) based on a new description of a $4^{q}$-QAM constellation, which aims to enlarge the family size of GCSs and obtain a low PMEPR. Furthermore, a previous construction of $4^{q}$-QAM GCSs presented by Li has been proved to be a special case of the new one, and the family size of new sequences is much larger than those previously mentioned, which means that there was a great improvement in the data rate. On the other hand, a previous construction of 16-QAM GCSs presented by Zeng is also a special case of the new one in this paper, when q=2. In the meantime, the proposed sequences have the same PMEPR upper bound as the previously mentioned sequences presented by Li when applied in OFDM systems, which increase the data rate without degrading the PMEPR performance. The theoretical analysis and simulation results show that the proposed new sequences can achieve a higher data rate and a low PMEPR.

## 1. Introduction

Orthogonal frequency division multiplexing (OFDM), as a high-performance physical layer technology of wireless communication, has been widely used in wireless communication systems for its advantages of high spectrum utilization, high power utilization, strong resistance to multipath delay expansion, and frequency-selective fading [1]. More specifically, IEEE 802.11 uses OFDM in wireless local area network (LAN) applications, and 802.16 uses OFDM in wireless network applications.

High-order modulations such as quadrature phase shift keying (QPSK), 16-quadrature amplitude modulation (QAM), 64-QAM, and so on, are generally applied in high-speed communication systems in order to acquire a higher data rate [2]. When it comes to OFDM, higher-order modulation will lead to a higher peak-to-mean envelope power ratio (PMEPR) of transmitted OFDM signals because an OFDM signal consists of many independently modulated subcarriers, which will cause a large PMEPR when coherently superimposed. A large PMEPR can cause a series of negative effects, such as increasing the complexity of analog-to-digital converters (ADCs) and digital-to-analog converters (DACs) and reducing the efficiency of radio frequency (RF) power amplifiers [3], which will lead to the serious degradation of the bit error ratio (BER) performance [4]. In addition, a large PMEPR can cause out-of-band radiation, which leads to severe inter-channel interference (ICI) [5]. Therefore, reducing the PMEPR is an effective means to improve the transmission efficiency and reduce the communication overhead of an OFDM system [6,7].

In order to obtain a lower PMEPR, Golay complementary sequences (GCSs) are usually considered in OFDM modulation [8]. In the meantime, QAM is diffusely applied in high-speed OFDM communication systems. Hence, QAM GCSs have been widely investigated in communication and signal processing research. In 1999, Davis and Jedwab [9] constructed a set of phase shift keying (PSK) Golay sequences using a direct non-recursive construction of polyphase complementary sequences based on generalized Boolean functions. The resulting Golay sequences were subsequently referred to as Golay–Davis–Jedwab (GDJ) sequences. In 2001, Rößing and Tarokh [10] proposed a new construction of 16-QAM GCSs by adding two QPSK GDJ sequences. They proved that the code rate of these sequences was twice that of the PSK Golay sequence, at the cost of a slightly higher PMEPR. Since then, many other constructions with a low PMEPR based on QAM constellations have been proposed [11,12,13,14,15,16].

In 2003, Tarokh and Sadjadpour [17] proposed the construction of $2^{2n}$-QAM sequences by using *n* QPSK GCSs, which was the extension of Rößing and Tarokh’s results in [10]. When applied in an OFDM system, the set of sequences had the PMEPR upper bounds of $6{\left(2^{n/2}-1\right)}^{2}/\left(2^{n}-1\right)$. In the same year, Chong et al. [18] proposed a new construction of 16-QAM GCSs by combining two QPSK GDJ sequences with one offset and a pair difference, which enlarged the family size to $\left(14+12m\right)\left(m!/2\right)4^{m+1}$ and bounded the PMEPR to 3.6. In 2006, Lee and Golomb [19] constructed new 64-QAM GCSs by combining three QPSK GDJ sequences with two offsets, which had the PMEPR bounded by 4.66. In 2008, Li [20] proposed some modifications and extensions for [18,19]. In 2009, Wang et al. [21] extended the construction of $2^{2n}$-QAM sequences with a new method and gave an upper bound on the PMEPR. In 2010, Chang et al. [22] constructed new 64-QAM GCSs and proved the conjecture presented in [19]. In addition, Li [23] proposed a new construction of $4^{q}$-QAM GCSs by combining *q* QPSK GDJ sequences with q−1 offsets and then calculated the family size. The PMEPR upper bounds of the proposed sequences in [23] have been proven to be $6\left(2^{q}-1\right)/\left(2^{q}+1\right)$, which approach six as the QAM constellation size increases. It can be noted that all previous QAM GCSs are special cases of the construction in [23] and are referred to as the generalized cases I-III. From the previous references, it can be seen that the family size of this general structure is mainly determined by the number of offsets, which implies one possible method of expanding the family size: expanding the number of offsets.

In 2013, Liu et al. [24] proposed constructions of new cases based on the nonsymmetrical Gaussian integer pair, which were different from the cases in [23] and were referred to as the generalized cases IV-V. Note that all the aforementioned constructions are based on the standard QPSK GDJ GCSs. In 2014, Zeng et al. [25] proposed a new construction of 16-QAM GCSs based on non-standard QPSK GDJ GCSs. In 2018, Zeng et al. [26] developed a new construction yielding $4^{q}$-QAM GCSs with a length of 2 m (integer m≥2), which included the known QAM GCSs with binary inputs as special cases and had the same PMEPR upper bound when used in an OFDM system. In the same year, Budišin and Spasojević [27] proposed a new recursive algorithm to generate a large number of standard QAM GCSs based on unitary matrices, and whose 1-Qum (based on one unitary matrix) case and 2-Qum (based on two unitary matrices) case could generate the QAM GCSs in the generalized cases I-V [23,24]. In 2019, Zeng et al. [28] proposed new constructions of 16-QAM GCSs, which further increased the family size and had the same PMEPR upper bound as the known sequences. In the same year, Zeng et al. [29] expanded their conclusion to $4^{q}$-QAM GCSs with the same methods, and the new sequences had the same PMEPR upper bounds as those mentioned above. In 2021, Wang et al. [30] provided a new method of constructing complementary sequence sets (CSSs) and complete complementary codes (CCCs) by using para-unitary (PU) matrices, which significantly increased the number of the sequences with a low PMEPR. In 2022, Wang et al. [31] proposed two new constructions of $4^{q}$-QAM GCSs by providing new compatible offsets based on the factorization of the integer *q*, which had the generalized cases I-V [23,24] as special cases. Since the family size directly determines the data rate, expanding the family size of sequences is an effective way to improve the data rate [32].

As the data rates and mobility supported by the OFDM system increase, the number of subcarriers also increases, resulting in a high PMEPR. However, reducing the PMEPR will increase the computational complexity. To solve this problem, many PMEPR reduction schemes that reduce the computational complexity of OFDM systems have been proposed. In 2013, Rahmatallah and Mohan [33] generated a taxonomy of the available solutions to mitigate the high PMEPR problem in OFDM systems. They also provided complexity analyses for several PMEPR reduction methods to demonstrate the differences in computational complexity between different methods. In 2017, Zhao et al. [34] proposed an improved joint optimization scheme, which combined the partial transmit sequence (PTS) and clipping and filtering (CF) methods with great PMEPR reduction performance. In the same year, Joo et al. [35] proposed two PTS schemes without side information (SI) for reducing the PMEPR of OFDM signals, which did not reduce the BER performance as compared to the conventional PTS with perfect SI. The above-mentioned PMEPR reduction schemes can effectively solve the problem and improve the performance of OFDM systems.

Based on the above literature search, it was found that increasing the family size is an effective way to improve the data rate, and the family size is mainly determined by the number of offsets in the general structure of QAM GCSs. This is the source of the research idea of this paper. The innovation points and main contributions of this paper are summarized as follows:

A new description of a $4^{q}$-QAM constellation with q+1 independent quaternary variables is presented in this paper, which has one more variable than the previous description and includes it as a special case;On this basis, a new construction of $4^{q}$-QAM GCSs is proposed, which greatly increases the family size and improves the data rate;More specifically, the new construction of the QAM sequences includes the construction in [23] as a special case and has a larger family size, which means a higher data rate;At the same time, the construction of 16-QAM GCSs in [28] is also a special case in this paper when q=2;Furthermore, the proposed sequences in this paper have the same PMEPR upper bounds as the known ones, which increase the data rate without degrading the PMEPR performance.

The rest of this paper is organized as follows. In Section 2, some background information is provided, including the definitions of GCSs and generalized Boolean functions (GBFs), the construction of QAM signals from the QPSK constellation, the PMEPR upper bound of GCSs, and some related conclusions. In Section 3, a new description of a $4^{q}$-QAM constellation is first presented. Based on this foundation, a new construction of $4^{q}$-QAM GCSs is proposed, and an example of 64-QAM GCSs is given to verify this conclusion. Then, the family size and the PMEPR upper bound of the new construction are described. In Section 4, the main results are summarized, and the main conclusions are given.

## 2. Materials and Methods

In this section, we provide some necessary materials, including the definitions of GCSs and GBFs, the construction of QAM signals from a QPSK constellation, and the PMEPR upper bound of the GCSs as well as several related lemmas.

### 2.1. Golay Complementary Sequences

Given two sequences of length N with complex elements, $A=\left[A_{0},A_{1},\cdots ,A_{N-1}\right]$ and $B=\left[B_{0},B_{1},\cdots ,B_{N-1}\right]$, we define [36]
(1)$$C_{A,B}\left(\tau \right)=\left\{\begin{array}{c} \overset{N-1-\tau }{\underset{i=0}{\sum }}A_{i}B_{i+\tau }^{*}\;\;0\leq \tau \leq N-1\\ \overset{N-1+\tau }{\underset{i=0}{\sum }}A_{i-\tau }B_{i}^{*}\;\;1-N\leq \tau <0\\ 0\;\;\;\;\;\;\left|\tau \right|\geq N \end{array}\right.$$
to be an aperiodic correlation function (ACF) of ***A*** and ***B***. More specifically, we say that $C_{A,A}\left(\tau \right)$ is an aperiodic autocorrelation function when A=B, which can be simplified as $C_{A}\left(\tau \right)$. If not, we call $C_{A,B}\left(\tau \right)$ an aperiodic cross-correlation function when A≠B [37].

For two sequences, ***A*** and ***B***, if they satisfy [38]
(2)$$C_{A}\left(\tau \right)+C_{B}\left(\tau \right)=0\;\left(\forall \tau \neq 0\right),$$
we say that (***A***, ***B***) forms a Golay complementary pair, and that ***A***, ***B*** are both Golay complementary sequences (GCSs).

### 2.2. Generalized Boolean Functions and Standard $2^{h}$-PSK GDJ GCSs

We define $Z_{2^{h}}=\left\{0,1,2,\cdots ,2^{h}-1\right\}\;\left(integer\;h\geq 1\right)$, and then the functions [9]
(3)$$f\left(x\right)=2^{h-1}\sum_{k=1}^{m-1}x_{\pi \left(k\right)}x_{\pi \left(k+1\right)}+\sum_{k=1}^{m}c_{k}x_{k}$$
are referred to as standard GBFs, where *π* stands for a permutation of the symbol set $\left\{1,2,\cdots ,m\right\}$, vector $x=\left(x_{1},x_{2},\cdots ,x_{m}\right)\in Z_{2}^{m}$, *m* is a positive integer that satisfies m≥2, and $c_{k}\in Z_{2^{h}}\;\left(1\leq k\leq m\right)$.

If the m-dimensional vector $x=\left(x_{1},x_{2},\cdots ,x_{m}\right)$ is preset, we can get a Boolean function value $f\left(x\right)\in Z_{2^{h}}$. It is obvious that the binary form of the integers from 0 to $2^{m}-1$ can be denoted by the vector ***x*** when it ranges from $\left(0,0,\cdots ,0\right)$ to $\left(1,1,\cdots ,1\right)$, and as a result, the $2^{m}$ function values in $Z_{2^{h}}$ can be produced [39].

On the basis of the standard GBFs, Davis and Jedwab discovered the connection between $2^{h}$-PSK GCSs and the generalized Reed–Muller codes; this led to the construction of a large class of PSK GCSs, which are well-known as the standard $2^{h}$-PSK GDJ GCSs [40]. Several related conclusions about these sequences are presented below.

**Lemma** **1**(Ref. [9], Corollary 4)**.**
*The Golay sequences over $Z_{2^{h}}$
of length $2^{m}$, determined by Equation (3), in total have $2^{h\left(m+1\right)}\cdot m!/2$.*

**Lemma** **2**(Ref. [9], Corollary 5)**.** Let


(4)
$$\left\{\begin{array}{c} a\left(x\right)=f\left(x\right)+c\\ b\left(x\right)=f\left(x\right)+2^{h-1}x_{\pi \left(1\right)}+c^{\prime }, \end{array}\right.\;$$



*where $c,\;c’\in Z_{2^{h}}$, and then the resultant sequences **a** and **b** are called a Golay complementary pair with length $2^{m}\;$over $Z_{2^{h}}$.*


### 2.3. Construction of QAM Signals from a QPSK Constellation

The QPSK constellation can be described based on the quaternary symbols $Z_{4}=\left\{0,1,2,3\right\}$ using the following set [11]:(5)$$\Omega_{QPSK}=\left\{j^{v}|v\in Z_{4}\right\}.$$

On the other hand, the $4^{q}$-QAM constellation (positive integer q≥2) can be described with the following set [29]:(6)$$\Omega_{4^{q}-QAM}=\left\{a+bj|-2^{q}+1\leq a,b\leq 2^{q}-1,a,b\;odd,j^{2}=-1\right\}.$$

One of the methods to produce a $4^{q}$-QAM constellation is through the use of QPSK symbols with the shift and rotation operations. Figure 1 shows the construction of a 64-QAM constellation by adding three QPSK symbols [23].

With the same method, the general $4^{q}$-QAM constellation can be expressed as [17]:(7)$$\Omega_{4^{q}-QAM}=\left\{\left(1+j\right)\sum_{p=0}^{q-1}2^{p}j^{v^{\left(p\right)}}|v^{\left(p\right)}\in Z_{4}\right\}.$$

When the *q*-dimensional vector $\left(v^{\left(0\right)},v^{\left(1\right)},\cdots ,v^{\left(q-1\right)}\right)$ varies from $\left(0,0,\cdots ,0\right)$ to $\left(3,3,\cdots ,3\right)$, the above equation correspond exactly to the $4^{q}$-QAM constellation.

### 2.4. Family Size and Code Rate

The family size of sequences directly determines the data rate. More specifically, when applied in OFDM systems, the family size affects the selection of the number of subcarriers. In [18], the definition of the code rate is provided, which is introduced to guide the selection of subcarriers. The following lemma gives the definition.

**Lemma** **3**(Ref. [18])**.**
*The code rate of a code C consisting of sequences of length N symbols is [18]*


(8)
$$R\left(C\right)=\frac{log_{2}\left|C\right|}{N},$$



*where $\left|C\right|$ stands for the family size of the code C.*


### 2.5. PMEPR Upper Bound of GCSs

Consider an OFDM system that has *N* subcarriers, $f_{0}$ is the carrier frequency, and $f_{i}$ is the frequency of the *i*th subcarrier, where $f_{i}=f_{0}+i\Delta f\;\left(0\leq i\leq N-1\right)$, and Δf is the bandwidth between each sub-channel; hence, the transmitted complex signal $S_{a}\left(t\right)$, which is encoded by the sequence $a=\left(a_{0},a_{1},\cdots ,a_{N-1}\right)$, is represented as follows [41]:(9)$$S_{a}\left(t\right)=\sum_{i=0}^{N-1}a_{i}e^{2\pi jf_{i}t}$$

Let *C* represent the ensemble of all possible codewords $\left(a\in C\right)$, and $p\left(a\right)$ indicate how likely the codeword ***a*** is to be transmitted. Thus, the average envelope power of the transmitted signal is written as follows [13]:(10)$$P_{av}=\sum_{a\in C}{\left|a\right|}^{2}p\left(a\right)$$

If the instantaneous envelope power of the transmitted OFDM signal is $P\left(t\right)={\left|S_{a}\left(t\right)\right|}^{2}$, then we write the PMEPR of the codeword ***a*** as [13]
(11)$$PMEPR\left(a\right)=\frac{\max\left(P\left(t\right)\right)}{P_{av}}.$$

The following lemma holds if an OFDM signal is encoded by binary, quaternary, or polyphase GCSs.

**Lemma** **4**(Ref. [9])**.**
*If a code C is made up of binary, quaternary, or polyphase GCSs, the PMEPR upper bound of the code C satisfies [9]*


(12)
$$PMEPR\left(C\right)\leq 2.$$


However, when it comes to $4^{q}$-QAM GCSs, which are proposed in [23], the following lemma gives their PMEPR upper bound.

**Lemma** **5**(Ref. [23])**.**
*If a code C is made up of $4^{q}$
-QAM GCSs constructed in [23], the PMEPR upper bound of the code C satisfies [23]*


(13)
$$PMEPR\left(C\right)\leq \frac{6\left(2^{q}-1\right)}{2^{q}+1}.$$


## 3. Results and Discussion

In this section, we first present a new description of a $4^{q}$-QAM constellation. On this basis, we propose a new construction of $4^{q}$-QAM GCSs and give an example of 64-QAM GCSs to verify this proposal. Then, we describe the family size and the PMEPR upper bound of the new construction.

### 3.1. New Description of $4^{q}$-QAM Constellation

In the description of the $4^{q}$-QAM constellation in Equation (7), there are *q* independent quaternary variables. Here is a new description of the $4^{q}$-QAM constellation presented by this paper, which is given by the following theorem:

**Theorem** **1.**

(14)
$$\Omega_{4^{q}-QAM}=\{\left(1+j\right)\left(\sum_{p=0}^{q-2}2^{p}j^{v^{\left(p\right)}}+2^{q-2}\sum_{p=q-1}^{q}j^{v^{\left(p\right)}}\right)|v^{\left(p\right)}\in Z_{4}\}$$

This new description has q+1 independent quaternary variables, which is one more than the description in Equation (7).

**Proof** **of** **Theorem** **1**([29])**.** We divide the proof into two parts: (1) each symbol in the set of Equation (14) must be included in the $4^{q}$-QAM constellation $\Omega_{4^{q}-QAM}$; (2) each $4^{q}$-QAM symbol in $\Omega_{4^{q}-QAM}$ can be produced by the set in Equation (14).

(1) For $\forall \left(v^{\left(0\right)},v^{\left(1\right)},\cdots ,v^{\left(q\right)}\right)\in Z_{4}^{q+1}$, whose symbol distribution of q+1 offsets are represented as
(15)$$\left\{\begin{array}{c} n_{0}=\left|\left\{p|v^{\left(p\right)}=0,\;0\leq p\leq q\right\}\right|\\ n_{1}=\left|\left\{p|v^{\left(p\right)}=1,\;0\leq p\leq q\right\}\right|\\ n_{2}=\left|\left\{p|v^{\left(p\right)}=2,\;0\leq p\leq q\right\}\right|\\ n_{3}=\left|\left\{p|v^{\left(p\right)}=3,\;0\leq p\leq q\right\}\right|, \end{array}\right.$$
the description of Equation (14) produces the following symbol:(16)$$\begin{array}{ll} S & =\left(1+j\right)\left(\overset{q-2}{\underset{p=0}{\sum }}2^{p}j^{v^{\left(p\right)}}+2^{q-2}\overset{q}{\underset{p=q-1}{\sum }}j^{v^{\left(p\right)}}\right)\\ & =\left(1+j\right)\left[\left(n_{0}-n_{2}\right)+j\left(n_{1}-n_{3}\right)\right]\\ & =\left(n_{0}-n_{1}-n_{2}+n_{3}\right)+j\left(n_{0}+n_{1}-n_{2}-n_{3}\right). \end{array}$$

It is obvious that $n_{0}+n_{1}+n_{2}+n_{3}=2^{q}-1$, which is an odd integer, so the values of the four integers $n_{0},n_{1},n_{2}$, and $n_{3}$ only have two cases: “one odd and three evens” or “one even and three odds”. Apparently, both cases have the same conclusion, which is that the values of the integers $n_{0}-n_{1}-n_{2}+n_{3}$ and $n_{0}+n_{1}-n_{2}-n_{3}$ are both odd. Based on this conclusion, we clearly have
(17)$$\begin{array}{c} -\left(2^{q}-1\right)\leq \end{array}n_{0}-n_{1}-n_{2}+n_{3},n_{0}+n_{1}-n_{2}-n_{3}\leq 2^{q}-1.$$

To sum up, we reach the following conclusion: $S\in \Omega_{4^{q}-QAM}$.

(2) For $\forall a+jb\in \Omega_{4^{q}-QAM}$, combined with Equation (14), we can obtain the following equation:(18)$$\sum_{p=0}^{q-2}2^{p}j^{v^{\left(p\right)}}+2^{q-2}\sum_{p=q-1}^{q}j^{v^{\left(p\right)}}=\frac{a+jb}{1+j}=\frac{a+b}{2}+j\frac{b-a}{2}$$

We then need to prove that there is at least one $\left(q+1\right)$-dimensional vector $\left(v^{\left(0\right)},v^{\left(1\right)},\cdots ,v^{\left(q\right)}\right)\in Z_{4}^{q+1}$ that satisfies Equation (18). We chose the vector by using the following strategy.

Step 1: We discretionarily chose $\left|\frac{a+b}{2}\right|$ “0s” or “2s” in this vector, depending on the sign of $\frac{a+b}{2}$.

Step 2: We discretionarily chose $\left|\frac{b-a}{2}\right|$ “1s” or “3s” in the remaining items aside from the chosen part in Step 1, depending on the sign of $\frac{b-a}{2}$.

Step 3: We discretionarily chose “0 and 2” or “1 and 3” in pairs in the remaining items aside from the chosen part in Steps 1 and 2, so that all the powers of *j* from these unused items add up to zero.

From the above steps, we can draw the following conclusion. For each symbol in $\Omega_{4^{q}-QAM}$, we can find at least one $\left(q+1\right)$-dimensional vector $\left(v^{\left(0\right)},v^{\left(1\right)},\cdots ,v^{\left(q\right)}\right)\in Z_{4}^{q+1}$ to ensure this $4^{q}$-QAM symbol can be generated by Equation (18).

Thus, summarizing the above, Theorem 1 has been proved. □

More specifically, if we set
(19)$$\left\{\begin{array}{c} v^{\left(p\right)}=\mu^{\left(p\right)},0\leq p\leq q-2\\ v^{\left(q-1\right)}=v^{\left(q\right)}=\mu^{\left(q-1\right)}, \end{array}\right.$$
then Equation (14) can be converted into
(20)$$\begin{array}{ll} \Omega_{4^{q}-QAM} & =\{\left(1+j\right)\left(\overset{q-2}{\underset{p=0}{\sum }}2^{p}j^{\mu^{\left(p\right)}}+2^{q-2}\cdot 2\cdot j^{\mu^{\left(q-1\right)}}\right)|\mu^{\left(p\right)}\in Z_{4}\}\\ & =\left\{\left(1+j\right)\overset{q-1}{\underset{p=0}{\sum }}2^{p}j^{\mu^{\left(p\right)}}|\mu^{\left(p\right)}\in Z_{4}\right\}. \end{array}$$

Obviously, Equation (20) is equivalent to Equation (7).

Therefore, Equation (7) is a special case of Theorem 1.

### 3.2. New Construction of $4^{q}$-QAM GCS

#### 3.2.1. Construction of New QAM Sequences

Based on the description of Equation (14), we propose a new construction of $4^{q}$-QAM GCSs in this section.

Theorem 2.
*If h=2 in Equation (3), then the obtained functions $f\left(x\right)$ are quaternary GBFs. Hence, we let*



(21)
$$\begin{array}{c} \left\{\begin{array}{c} a^{\left(0\right)}\left(x\right)=f\left(x\right)+c\\ b^{\left(0\right)}\left(x\right)=a^{\left(0\right)}\left(x\right)+\mu \left(x\right)\\ a^{\left(1\right)}\left(x\right)=a^{\left(0\right)}\left(x\right)+s^{\left(1\right)}\left(x\right)\\ b^{\left(1\right)}\left(x\right)=a^{\left(1\right)}\left(x\right)+\mu \left(x\right)=a^{\left(0\right)}\left(x\right)+s^{\left(1\right)}\left(x\right)+\mu \left(x\right)\\ \vdots \\ a^{\left(q\right)}\left(x\right)=a^{\left(0\right)}\left(x\right)+s^{\left(q\right)}\left(x\right)\\ b^{\left(q\right)}\left(x\right)=a^{\left(q\right)}\left(x\right)+\mu \left(x\right)=a^{\left(0\right)}\left(x\right)+s^{\left(q\right)}\left(x\right)+\mu \left(x\right). \end{array}\right. \end{array}\;$$



*By means of Equation (14), the $4^{q}$-QAM sequences $A=\left(A_{0},A_{1},\cdots ,A_{N-1}\right)$ and $B=\left(B_{0},B_{1},\cdots ,B_{N-1}\right)$ with length $N=2^{m}$ can be constructed as follows:*

(22)
$$\left\{\begin{array}{c} A_{i}=\left(1+j\right)\left(\overset{q-2}{\underset{p=0}{\sum }}2^{p}j^{a_{i}^{\left(p\right)}}+2^{q-2}\overset{q}{\underset{p=q-1}{\sum }}j^{a_{i}^{\left(p\right)}}\right)\\ B_{i}=\left(1+j\right)\left(\overset{q-2}{\underset{p=0}{\sum }}2^{p}j^{b_{i}^{\left(p\right)}}+2^{q-2}\overset{q}{\underset{p=q-1}{\sum }}j^{b_{i}^{\left(p\right)}}\right), \end{array}\right.$$

*where $a_{i}^{\left(p\right)},b_{i}^{\left(p\right)}\in Z_{4}$, 0≤p≤q, $0\leq i\leq 2^{m}-1$. Then, the obtained $4^{q}$-QAM sequences A and B are $4^{q}$-QAM GCSs when the offsets $s^{\left(p\right)}\left(x\right)\;\left(1\leq p\leq q\right)$ and the corresponding pairing difference $\mu \left(x\right)$ with $d_{l}^{\left(p\right)}\in Z_{4}\;\left(1\leq p\leq q,0\leq l\leq 2\right)$ are as in one of the following cases:*

(23)
$$Case\;I:\;\left\{\begin{array}{c} \;\mu \left(x\right)=2x_{\pi \left(m\right)}\\ s^{\left(p\right)}\left(x\right)=d_{0}^{\left(p\right)}+d_{1}^{\left(p\right)}x_{\pi \left(1\right)}\;\left(1\leq p\leq q\right) \end{array}\right.$$


(24)
$$Case\;II:\;\left\{\begin{array}{c} \;\mu \left(x\right)=2x_{\pi \left(1\right)}\\ s^{\left(p\right)}\left(x\right)=d_{0}^{\left(p\right)}+d_{1}^{\left(p\right)}x_{\pi \left(m\right)}\;\left(1\leq p\leq q\right) \end{array}\right.$$


(25)
$$Case\;III:\;\left\{\begin{array}{c} \;\mu \left(x\right)=2x_{\pi \left(1\right)}\;or\;2x_{\pi \left(m\right)}\\ s^{\left(p\right)}\left(x\right)=d_{0}^{\left(p\right)}+d_{1}^{\left(p\right)}x_{\pi \left(\omega \right)}+d_{2}^{\left(p\right)}x_{\pi \left(\omega +1\right)}\\ with\;2d_{0}^{\left(p\right)}+d_{1}^{\left(p\right)}+d_{2}^{\left(p\right)}=0\;\left(mod\;4\right)\\ \left(1\leq p\leq q,1\leq \omega \leq m-1\right). \end{array}\right.$$



Particularly, if $s^{\left(q-1\right)}\left(x\right)=s^{\left(q\right)}\left(x\right)$, we have
(26)$$\left\{\begin{array}{c} A_{i}=\left(1+j\right)\overset{q-1}{\underset{p=0}{\sum }}2^{p}j^{a_{i}^{\left(p\right)}}\\ B_{i}=\left(1+j\right)\overset{q-1}{\underset{p=0}{\sum }}2^{p}j^{b_{i}^{\left(p\right)}}, \end{array}\right.$$
which is the same as the construction of Theorem 2 in [23].

Therefore, the construction of Theorem 2 in [23] is a special case of the one constructed by Theorem 2 in this paper. Figure 2 clearly depicts the process of how to construct the QAM GCSs from the QAM constellation and shows the relationship between [23] and this paper.

**Proof** **of** **Theorem** **2.**For ∀τ>0, the aperiodic autocorrelation function of the sequence ***A*** can be expressed as follows:



(27)
$$C_{A}\left(\tau \right)=\sum_{i=0}^{N-1-\tau }A_{i}A_{i+\tau }^{*}.$$



Combined with Equation (22), we can calculate Equation (27) into
(28)$$\begin{array}{ll} \frac{1}{2}C_{A}\left(\tau \right) & =\sum_{i=0}^{N-1-\tau }\begin{array}{l} {}[\left(\sum_{p=0}^{q-2}2^{p}j^{a_{i}^{\left(p\right)}}+2^{q-2}\sum_{p=q-1}^{q}j^{a_{i}^{\left(p\right)}}\right)\\ \cdot {\left(\sum_{p=0}^{q-2}2^{p}j^{a_{i+\tau }^{\left(p\right)}}+2^{q-2}\sum_{p=q-1}^{q}j^{a_{i+\tau }^{\left(p\right)}}\right)}^{*}] \end{array}\\ & =\sum_{p=0}^{q-2}4^{p}C_{a^{\left(p\right)}}\left(\tau \right)+4^{q-2}\sum_{p=q-1}^{q}C_{a^{\left(p\right)}}\left(\tau \right)\\ & +\sum_{\begin{array}{c} p^{\prime },p^{\prime \prime }=0\\ p^{\prime }\neq p^{\prime \prime } \end{array}}^{q-2}2^{p^{\prime }+p^{\prime \prime }}\left[C_{a^{\left(p^{\prime }\right)},a^{\left(p^{\prime \prime }\right)}}\left(\tau \right)+C_{a^{\left(p^{\prime \prime }\right)},a^{\left(p^{\prime }\right)}}\left(\tau \right)\right]\\ & +\sum_{\begin{array}{c} 0\leq p^{\prime }\leq q-2\\ q-1\leq p^{’’}\leq q \end{array}}2^{p^{\prime }+q-2}\left[C_{a^{\left(p^{\prime }\right)},a^{\left(p^{\prime \prime }\right)}}\left(\tau \right)+C_{a^{\left(p^{\prime \prime }\right)},a^{\left(p^{\prime }\right)}}\left(\tau \right)\right]\\ & +\sum_{\begin{array}{c} p^{\prime },p^{\prime \prime }=q-1\\ p^{\prime }\neq p^{\prime \prime } \end{array}}^{q}\left[C_{a^{\left(p^{\prime }\right)},a^{\left(p^{\prime \prime }\right)}}\left(\tau \right)+C_{a^{\left(p^{\prime \prime }\right)},a^{\left(p^{\prime }\right)}}\left(\tau \right)\right] \end{array}$$

For the sequence ***B*** in Equation (22), using the same method, we can get


(29)
$$\begin{array}{cc} \frac{1}{2}C_{B}\left(\tau \right) & =\overset{q-2}{\underset{p=0}{\sum }}4^{p}C_{a^{\left(p\right)}}\left(\tau \right)+4^{q-2}\overset{q}{\underset{p=q-1}{\sum }}C_{a^{\left(p\right)}}\left(\tau \right)\\ & +\overset{q-2}{\underset{\begin{array}{c} p^{\prime },p^{\prime \prime }=0\\ p^{\prime }\neq p^{\prime \prime } \end{array}}{\sum }}2^{p^{\prime }+p^{\prime \prime }}\left[C_{a^{\left(p^{\prime }\right)},a^{\left(p^{\prime \prime }\right)}}\left(\tau \right)+C_{a^{\left(p^{\prime \prime }\right)},a^{\left(p^{\prime }\right)}}\left(\tau \right)\right]\\ & +\underset{\begin{array}{c} 0\leq p^{\prime }\leq q-2\\ q-1\leq p^{’’}\leq q \end{array}}{\sum }2^{p^{\prime }+q-2}\left[C_{a^{\left(p^{\prime }\right)},a^{\left(p^{\prime \prime }\right)}}\left(\tau \right)+C_{a^{\left(p^{\prime \prime }\right)},a^{\left(p^{\prime }\right)}}\left(\tau \right)\right]\\ & +\overset{q}{\underset{\begin{array}{c} p^{\prime },p^{\prime \prime }=q-1\\ p^{\prime }\neq p^{\prime \prime } \end{array}}{\sum }}\left[C_{a^{\left(p^{\prime }\right)},a^{\left(p^{\prime \prime }\right)}}\left(\tau \right)+C_{a^{\left(p^{\prime \prime }\right)},a^{\left(p^{\prime }\right)}}\left(\tau \right)\right] \end{array}$$


By employing Lemma 2, we can get that sequences $a^{\left(p\right)}$ and $b^{\left(p\right)}$ $\left(0\leq p\leq q\right)$ form GCSs, and then we can obtain
(30)$$C_{a^{\left(p\right)}}\left(\tau \right)+C_{b^{\left(p\right)}}\left(\tau \right)=0\;(\forall \tau >0,\;0\leq p\leq q).$$

Hence, in order to ensure that sequences ***A*** and ***B*** are GCSs, the following equation would be a sufficient condition [41]:(31)$$\begin{array}{c} C_{a^{\left(p^{\prime }\right)},a^{\left(p^{\prime \prime }\right)}}\left(\tau \right)+C_{a^{\left(p^{\prime \prime }\right)},a^{\left(p^{\prime }\right)}}\left(\tau \right)+C_{b^{\left(p^{\prime }\right)},b^{\left(p^{\prime \prime }\right)}}\left(\tau \right)+C_{b^{\left(p^{\prime \prime }\right)},b^{\left(p^{\prime }\right)}}\left(\tau \right)=0\\ \left(\forall \tau >0,\;0\leq p^{\prime },p^{\prime \prime }\leq q,p^{\prime }\neq p^{\prime \prime }\right). \end{array}$$

There are two QPSK GCS pairs involved in Equation (31), represented as $\left(a^{p^{\prime }},b^{p^{\prime }}\right)$ and $\left(a^{p^{\prime \prime }},b^{p^{\prime \prime }}\right)$. Then, we have the following equation:(32)$$\begin{array}{c} \left\{\begin{array}{c} a^{\left(p^{\prime }\right)}\left(x\right)=f\left(x\right)+c\\ b^{\left(p^{\prime }\right)}\left(x\right)=a^{\left(p^{\prime }\right)}\left(x\right)+\mu \left(x\right)\\ a^{\left(p^{\prime \prime }\right)}\left(x\right)=a^{\left(p^{\prime }\right)}\left(x\right)+s\left(x\right)\\ b^{\left(p^{\prime \prime }\right)}\left(x\right)=b^{\left(p^{\prime }\right)}\left(x\right)+s\left(x\right)=a^{\left(p^{\prime \prime }\right)}+\mu \left(x\right). \end{array}\right. \end{array}$$

Let $i=\left(i_{1},i_{2},\cdots ,i_{m}\right)$ denote the binary representation of *i*, i.e., $i=\sum_{k=1}^{m}i_{k}2^{m-k}$.

Let $f_{i}$, $a_{i}$, $b_{i}$, $\mu_{i}$, and $s_{i}$ denote the *i*th elements of the sequences generated from $f\left(x\right)$, $a\left(x\right)$, $b\left(x\right)$, $\mu \left(x\right)$, and $s\left(x\right)$ over $Z_{4}$. From Equation (32), we have
(33)$$\left\{\begin{array}{c} a_{i}{}^{\left(p^{\prime }\right)}=f_{i}+c\\ b_{i}{}^{\left(p^{\prime }\right)}=a_{i}{}^{\left(p^{\prime }\right)}+\mu_{i}\\ a_{i}{}^{\left(p^{\prime \prime }\right)}=a_{i}{}^{\left(p^{\prime }\right)}+s_{i}\\ b_{i}{}^{\left(p^{\prime \prime }\right)}=b_{i}{}^{\left(p^{\prime }\right)}+s_{i}=a_{i}{}^{\left(p^{\prime \prime }\right)}+\mu_{i}. \end{array}\right.$$

Therefore,
(34)$$\begin{array}{l} \;C_{a^{\left(p^{\prime }\right)},a^{\left(p^{\prime \prime }\right)}}\left(\tau \right)+C_{a^{\left(p^{\prime \prime }\right)},a^{\left(p^{\prime }\right)}}\left(\tau \right)+C_{b^{\left(p^{\prime }\right)},b^{\left(p^{\prime \prime }\right)}}\left(\tau \right)+C_{b^{\left(p^{\prime \prime }\right)},b^{\left(p^{\prime }\right)}}\left(\tau \right)\\ =\sum_{i=0}^{N-1-\tau }\left(j^{a_{i}{}^{\left(p^{\prime }\right)}-a_{i+\tau }{}^{\left(p^{\prime \prime }\right)}}+j^{a_{i}{}^{\left(p^{\prime \prime }\right)}-a_{i+\tau }{}^{\left(p^{\prime }\right)}}+j^{b_{i}{}^{\left(p^{\prime }\right)}-b_{i+\tau }{}^{\left(p^{\prime \prime }\right)}}+j^{b_{i}{}^{\left(p^{\prime \prime }\right)}-b_{i+\tau }{}^{\left(p^{\prime }\right)}}\right)\\ =\sum_{i=0}^{N-1-\tau }\left[j^{a_{i}{}^{\left(p^{\prime }\right)}-a_{i+\tau }{}^{\left(p^{\prime }\right)}}\left(j^{-s_{i+\tau }}+j^{s_{i}}\right)+j^{b_{i}{}^{\left(p^{\prime }\right)}-b_{i+\tau }{}^{\left(p^{\prime }\right)}}\left(j^{-s_{i+\tau }}+j^{s_{i}}\right)\right]\\ =\sum_{i=0}^{N-1-\tau }\left[j^{a_{i}{}^{\left(p^{\prime }\right)}-a_{i+\tau }{}^{\left(p^{\prime }\right)}}\left(j^{-s_{i+\tau }}+j^{s_{i}}\right)+j^{a_{i}{}^{\left(p^{\prime }\right)}-a_{i+\tau }{}^{\left(p^{\prime }\right)}}j^{\mu_{i}-\mu_{i+\tau }}\left(j^{-s_{i+\tau }}+j^{s_{i}}\right)\right]\\ =\sum_{i=0}^{N-1-\tau }\left[j^{a_{i}{}^{\left(p^{\prime }\right)}-a_{i+\tau }{}^{\left(p^{\prime }\right)}}\left(j^{-s_{i+\tau }}+j^{s_{i}}\right)\left(1+j^{\mu_{i}-\mu_{i+\tau }}\right)\right]. \end{array}$$

The last summation in Equation (34) was verified to equal its own negation in [23], so Equation (34) must be zero. From this conclusion, it can be concluded that Equation (31) is valid.

Because Equation (31) was proved to be true, we can get
(35)$$C_{A}\left(\tau \right)+C_{B}\left(\tau \right)=0\;\left(\forall \tau >0\right),$$
which can prove that sequences ***A*** and ***B*** are GCSs, and then the proof of Theorem *2* is complete. □

An example is given below to verify this conclusion and make it easier for readers to understand.

In Theorem 1, let q=3, then a construction of 64-QAM is given by
(36)$$\Omega_{64-\mathrm{QAM}}=\left\{\left(1+j\right)\left(j^{v\left(0\right)}+2j^{v\left(1\right)}+2j^{v\left(2\right)}+2j^{v\left(3\right)}\right)\left|v^{\left(0\right)},v^{\left(1\right)},v^{\left(2\right)},v^{\left(3\right)}\in Z_{4}\right.\right\}.$$

Then, the 64-QAM sequences $A=\left(A_{0},A_{1},\cdots ,A_{15}\right)$ and $B=\left(B_{0},B_{1},\cdots ,B_{15}\right)$ with length $N=2^{4}=16$ can be constructed as
(37)$$\left\{\begin{array}{c} A_{i}=\left(1+j\right)\left(j^{a_{i}^{\left(0\right)}}+2j^{a_{i}^{\left(1\right)}}+2j^{a_{i}^{\left(2\right)}}+2j^{a_{i}^{\left(3\right)}}\right)\\ B_{i}=\left(1+j\right)\left(j^{b_{i}^{\left(0\right)}}+2j^{b_{i}^{\left(1\right)}}+2j^{b_{i}^{\left(2\right)}}+2j^{b_{i}^{\left(3\right)}}\right). \end{array}\right.$$

In Theorem *2*, by employing *Case I*, let $x=\left(x_{1},x_{2},x_{3},x_{4}\right)$, the standard GBF $f\left(x\right)=2\left(x_{1}x_{2}+x_{2}x_{3}+x_{3}x_{4}\right)+x_{1}+3x_{3}$, the offsets $s^{\left(1\right)}\left(x\right)=3$, $s^{\left(2\right)}\left(x\right)=1+x_{1}$, $s^{\left(3\right)}\left(x\right)=3x_{1}$, and the pairing difference $\mu \left(x\right)=2x_{4}$; then, we can get
(38)$$\begin{array}{c} \left\{\begin{array}{c} a^{\left(0\right)}\left(x\right)=2\left(x_{1}x_{2}+x_{2}x_{3}+x_{3}x_{4}\right)+x_{1}+3x_{3}\\ b^{\left(0\right)}\left(x\right)=a^{\left(0\right)}\left(x\right)+2x_{4}\\ a^{\left(1\right)}\left(x\right)=a^{\left(0\right)}\left(x\right)+3\\ b^{\left(1\right)}\left(x\right)=a^{\left(1\right)}\left(x\right)+2x_{4}\\ a^{\left(2\right)}\left(x\right)=a^{\left(0\right)}\left(x\right)+1+x_{1}\\ b^{\left(2\right)}\left(x\right)=a^{\left(2\right)}\left(x\right)+2x_{4}\\ a^{\left(3\right)}\left(x\right)=a^{\left(0\right)}\left(x\right)+3x_{1}\\ b^{\left(3\right)}\left(x\right)=a^{\left(3\right)}\left(x\right)+2x_{4}. \end{array}\right. \end{array}$$

Therefore, the proposed 64-QAM sequences with a length of 16 are represented as
(39)$$\begin{array}{l} \left\{\begin{array}{l} A=\left(3+3j,3+3j,3-3j,-3+3j,3+3j,3+3j,-3+3j,3-3j,\right.\\ \;\left.5+3j,5+3j,3-5j,-3+5j,-5-3j,-5-3j,3-5j,-3+5j\right)\\ B=\left(3+3j,-3-3j,3-3j,3-3j,\right.3+3j,-3-3j,-3+3j,-3+3j,\\ \;\left.5+3j,-5-3j,3-5j,3-5j,-5-3j,5+3j,3-5j,3-5j\right). \end{array}\right. \end{array}$$

Consequently, the sum of their autocorrelation function is
(40)$$C_{A}\left(\tau \right)+C_{B}\left(\tau \right)=\left(832,0,0,0,0,0,0,0,0,0,0,0,0,0,0,0,0,0,0,0,0,0,0,0,0,0,0,0,0,0,0\right).$$

Obviously, the results satisfy $C_{A}\left(\tau \right)+C_{B}\left(\tau \right)=0\;\left(\forall \tau \neq 0\right)$, which means sequences ***A*** and ***B*** are both 64-QAM GCSs.

Sequence ***A***’s autocorrelation function $C_{A}\left(\tau \right)$, sequence ***B***’s autocorrelation function $C_{B}\left(\tau \right)$, and their sum $C_{A}\left(\tau \right)+C_{B}\left(\tau \right)$ were calculated, and the results are depicted in Figure 3.

#### 3.2.2. Family Size of New QAM Sequences

Combined with Equation (21), we can convert Equation (22) into
(41)$$\left\{\begin{array}{c} A_{i}=\left(1+j\right)\left(\overset{q-2}{\underset{p=0}{\sum }}2^{p}j^{a_{i}^{\left(0\right)}+s_{i}^{\left(p\right)}}+2^{q-2}\overset{q}{\underset{p=q-1}{\sum }}j^{a_{i}^{\left(0\right)}+s_{i}^{\left(p\right)}}\right)\\ B_{i}=\left(1+j\right)\left(\overset{q-2}{\underset{p=0}{\sum }}2^{p}j^{a_{i}^{\left(0\right)}+s_{i}^{\left(p\right)}+\mu_{i}}+2^{q-2}\overset{q}{\underset{p=q-1}{\sum }}j^{a_{i}^{\left(0\right)}+s_{i}^{\left(p\right)}+\mu_{i}}\right). \end{array}\right.$$

Apparently, if the offset vector $\left(s^{\left(1\right)}\left(x\right),s^{\left(2\right)}\left(x\right),\cdots ,s^{\left(q-1\right)}\left(x\right),s^{\left(q\right)}\left(x\right)\right)$ is resolved, then the family size of the $4^{q}$-QAM GCSs pair (***A***, ***B***) is determined. Notice two arbitrary offset vectors $\left(g^{\left(1\right)}\left(x\right),g^{\left(2\right)}\left(x\right),\cdots ,g^{\left(q-1\right)}\left(x\right),g^{\left(q\right)}\left(x\right)\right)$ and $\left(h^{\left(1\right)}\left(x\right),h^{\left(2\right)}\left(x\right),\cdots ,h^{\left(q-1\right)}\left(x\right),\;h^{\left(q\right)}\left(x\right)\right)$; if they satisfy $g^{\left(k\right)}\left(x\right)=h^{\left(k\right)}\left(x\right)\;\left(1\leq k\leq q-2\right)$, $g^{\left(q-1\right)}\left(x\right)=h^{\left(q\right)}\left(x\right),$ and $g^{\left(q\right)}\left(x\right)=h^{\left(q-1\right)}\left(x\right)$, then these two offset vectors produce the same two $4^{q}$-QAM GCS pairs. To avoid the above case of a repeated count, we considered the two cases below to calculate the family size.

*Case I:*$s^{\left(q-1\right)}\left(x\right)=s^{\left(q\right)}\left(x\right)$.

In this case, our construction was just the same as the one in [23]. According to the conclusion of Corollary 3 in [23], we can obtain from $\left(m+1\right)\cdot 4^{2\left(q-1\right)}-\left(m+1\right)\cdot 4^{q-1}+2^{q-1}$ different offset vectors.

*Case II:*$s^{\left(q-1\right)}\left(x\right)\neq s^{\left(q\right)}\left(x\right)$.

In this case, we divided the offset vector $\left(s^{\left(1\right)}\left(x\right),s^{\left(2\right)}\left(x\right),\cdots ,s^{\left(q-1\right)}\left(x\right),s^{\left(q\right)}\left(x\right)\right)$ into two parts. Part 1 is the front q−2 elements forming the $\left(q-2\right)$-dimensional offset vector $\left(s^{\left(1\right)}\left(x\right),s^{\left(2\right)}\left(x\right),\cdots ,\;s^{\left(q-2\right)}\left(x\right)\right)$, and Part 2 is the offset pair $\left(s^{\left(q-1\right)}\left(x\right),s^{\left(q\right)}\left(x\right)\right)$ consisting of the last two elements. Then, we calculated these two parts respectively and multiplied the results.

Part 1: Consider the offset vector $\left(s^{\left(1\right)}\left(x\right),s^{\left(2\right)}\left(x\right),\cdots ,\;s^{\left(q-2\right)}\left(x\right)\right)\;\left(s^{\left(p\right)}\left(x\right)\in Z_{4},1\leq p\leq q-2\right)$. When this $\left(q-2\right)$-dimensional vector ranges from $\left(0,0,\cdots ,0\right)$ to $\left(1,1,\cdots ,1\right)$, $4^{q-2}$ distinct offset vectors can be produced.

Part 2: Consider the offset pair $\left(s^{\left(q-1\right)}\left(x\right),s^{\left(q\right)}\left(x\right)\right)\;\left(s^{\left(p\right)}\left(x\right)\in Z_{4},q-1\leq p\leq q\right)$. By using the same method in [18], we can group the possible offsets into five groups, which satisfy the empty pairwise intersections, as below. It is known that the permutations of the offset coefficients $\left(d_{0}^{\left(p\right)},d_{1}^{\left(p\right)},d_{2}^{\left(p\right)}\right)$ satisfy $2d_{0}^{\left(p\right)}+d_{1}^{\left(p\right)}+d_{2}^{\left(p\right)}=0\;\left(mod\;4\right)$, as listed in Table 1 [23], and we can get
(42)$$\left\{\begin{array}{l} S_{1}=\{d_{0}|d_{0}=0,1,2,3\}\\ S_{2}=\{d_{0}+d_{1}x_{\pi \left(1\right)}|d_{0},d_{1}\in Z_{4},d_{1}\neq 0\}\\ S_{3}=\left\{d_{0}+d_{1}x_{\pi \left(m\right)}|d_{0},d_{1}\in Z_{4},d_{1}\neq 0\right\}\\ S_{4}=\text{\{}d_{0}+d_{1}x_{\pi \left(\omega \right)}|(d_{0},d_{1})=\left(1,2\right),\left(3,2\right)\}\;\left(2\leq \omega \leq m-1\right)\\ S_{5}=\text{\{}d_{0}+d_{1}x_{\pi \left(\omega \right)}+d_{2}x_{\pi \left(\omega +1\right)}|(d_{0},d_{1},d_{2})=\left(0,1,3\right),\left(0,2,2\right),\\ \;\left(0,3,1\right),\left(1,1,1\right),\left(1,3,3\right),\left(2,1,3\right),\left(2,2,2\right),\left(2,3,1\right),\\ \;\left(3,1,1\right),\left(3,3,3\right)\}\;\left(1\leq \omega \leq m-1\right). \end{array}\right.$$

In order to receive different offset pairs $\left(s^{\left(q-1\right)}\left(x\right),s^{\left(q\right)}\left(x\right)\right)$, we used the selection strategy presented in [29] to select the offset pairs in $S_{i}\;\left(1\leq i\leq 5\right)$. Step 1: Arbitrarily select an offset (expressed as $\Psi_{1}$) in $S_{i}$ as $s^{\left(q-1\right)}\left(x\right)$, then arbitrarily select an offset in $S_{i}-\Psi_{1}$ as $s^{\left(q\right)}\left(x\right)$. Apparently, we can obtain $\left|S_{i}\right|-1$ possible offset pairs in this step. Step 2: Arbitrarily select an offset (expressed as $\Psi_{2}$) in $S_{i}-\Psi_{1}$ as $s^{\left(q-1\right)}\left(x\right)$, then arbitrarily select an offset in $S_{i}-\Psi_{1}-\Psi_{2}$ as $s^{\left(q\right)}\left(x\right)$, which can produce $\left|S_{i}\right|-2$ possible offset pairs. Then, the above steps are repeated until we get $\left|S_{i}-\Psi_{1}-\Psi_{2}-\cdots \right|\leq 1$. Summarizing the results of the above steps, we can obtain $\left(\left|S_{i}\right|-1\right)+\left(\left|S_{i}\right|-2\right)+\cdots +1$ possible offset pairs in total.

Employing the above strategy, (a) for $S_{1}$, we calculated the possible offset pairs with 3+2+1=6; (b) for $S_{2}$, we calculated the possible offset pairs with 11+10+⋯+1=66; (c) for $S_{3}$, this used the same situation as the previous one; (d) for $S_{4}$, there existed only one offset pair. Furthermore, the parameter ω can vary from 2 to m−1, so the number of possible offset pairs is m−2 in total; (e) for $S_{5}$, we calculated the possible offset pairs with 9+8+⋯+1=45. In addition, the parameter ω can vary from 1 to m−1 in each offset pair. Hence, there are a total of $45\left(m-1\right)$ possible offset pairs in this case. By adding up (a) to (e), we can obtain 46m+91 possible offset pairs in total.

Combining Part 1 and Part 2, *Case II* has a total of $\left(46m+91\right)\cdot 4^{q-2}$ possible offset pairs.

By summing all the possible offset pairs in *Case I* and *Case II*, the results show that there are $\left(46m+91\right)\cdot 4^{q-2}+\left(m+1\right)\cdot 4^{2\left(q-1\right)}-\left(m+1\right)\cdot 4^{q-1}+2^{q-1}$ different offset pairs in total. Thus, by employing Lemma 1, the theorem below gives the family size of the new QAM sequences.

**Theorem** **3.**Consider the $4^{q}$
-QAM GCSs of length $2^{m}$ constructed by Theorem 1. Thus, the number of the sequences is


(43)
$$\begin{array}{c} \left[\left(46m+91\right)\cdot 4^{q-2}+\left(m+1\right)\cdot 4^{2\left(q-1\right)}-\left(m+1\right)\cdot 4^{q-1}+2^{q-1}\right]\\ \cdot \left(m!/2\right)4^{m+1}\;\left(m\geq 2,q\geq 2\right). \end{array}$$


More specifically, let q=2 in Equation (43), and we can obtain that the number of 16-QAM GCSs is
(44)$$\left(58m+105\right)\cdot \left(m!/2\right)4^{m+1}\;\left(m\geq 2\right),$$
which is exactly the result of Theorem 6 in [28]. Hence, the conclusion of [28] is a special case in this paper when q=2.

Figure 4 depicts the comparison of the family sizes of 16-QAM GCSs when q=2 between [23] and this paper; it can be seen that the number of sequences increased significantly. Table 2 shows the comparison of the code rates and family sizes between [23,28] and this paper, providing a visual representation of the data rate improvement.

#### 3.2.3. PMEPR Upper Bound of New QAM Sequences

The PMEPR upper bound of the new $4^{q}$-QAM GCSs are represented by the following theorem.

**Theorem** **4.**
*Consider a code C whose codewords are made up of $4^{q}$
-QAM GCSs constructed by Theorem 2, then the PMEPR upper bound of the code C satisfies*



(45)
$$PMEPR\left(C\right)\leq \frac{6\left(2^{q}-1\right)}{2^{q}+1}.$$


**Proof** **of** **Theorem** **4.**For ∀A∈C, let ***A*** be a $4^{q}$-QAM GCSs with length N constructed by Theorem 2, then the peak envelope power (PEP) of sequence ***A*** satisfies $PEP\left(A\right)\leq 2\overset{N-1}{\underset{i=0}{\sum }}{\left|A_{i}\right|}^{2}$ due to Equation (26) in [23]. Combined with Equation (22), and in order to ensure that the average squared magnitude is equal to one, we have


(46)
$$\begin{array}{c} {\left|\frac{1}{\sqrt{\left(4^{q}-1\right)/3}}A_{i}\right|}^{2}={\left|\frac{1+j}{\sqrt{\left(4^{q}-1\right)/3}}\left(\overset{q-2}{\underset{p=0}{\sum }}2^{p}j^{a_{i}^{\left(p\right)}}+2^{q-2}\overset{q}{\underset{p=q-1}{\sum }}j^{a_{i}^{\left(p\right)}}\right)\right|}^{2}\\ \leq \frac{{\left(2^{q}-1\right)}^{2}}{\left(4^{q}-1\right)/3}=\frac{3\left(2^{q}-1\right)}{2^{q}+1}. \end{array}$$


Then, we can get
(47)$$\begin{array}{c} PMEPR\left(A\right)=\frac{PEP\left(A\right)}{N}\leq \frac{2\sum_{i=0}^{N-1}{\left|A_{i}\right|}^{2}}{N}\\ \;\leq \frac{2N}{N}\cdot \frac{3\left(2^{q}-1\right)}{2^{q}+1}=\frac{6\left(2^{q}-1\right)}{2^{q}+1}, \end{array}$$
and the proof is complete. □

The PMEPR upper bound is equal to 3.6 when q=2 (16-QAM), equal to 4.667 when q=3 (64-QAM), and equal to 5.294 when q=4 (256-QAM), and it approaches 6 as the constellation order increases [42]. Figure 5 clearly depicts this tendency.

As a result, the new QAM GCSs in this paper have the same PMEPR upper bound as the sequences in [23], which means that there was no degradation in the PMEPR performance.

## 4. Conclusions

In this paper, a new description of a $4^{q}$-QAM constellation has been presented. Based on this foundation, we proposed a new construction of $4^{q}$-QAM GCSs of length $2^{m}$ using GBFs, which resulted in the enlargement of the family size of GCSs and allowed us to obtain a low PMEPR. A previous construction of $4^{q}$-QAM GCSs presented by Li and another previous construction of 16-QAM GCSs presented by Zeng were proven to be special cases of ours. The family size of the new sequences was calculated to be $\left[\left(46m+91\right)\cdot 4^{q-2}+\left(m+1\right)\cdot 4^{2\left(q-1\right)}-\left(m+1\right)\cdot 4^{q-1}+2^{q-1}\right]\cdot \left(m!/2\right)4^{m+1}$. This result shows that the new sequences have a larger family size, which means that there was a great improvement in the data rate. When applied in OFDM systems, the new sequences have the same PMEPR upper bound of $6\left(2^{q}-1\right)/\left(2^{q}+1\right)$ as the sequences presented by Li, which means we increased the data rate without degrading the PMEPR performance. Our next research directions will be to propose the PMEPR reduction schemes by reducing the computational complexity of OFDM systems, and to focus on the future challenges of a lower PMEPR by improving or reducing the computational complexity of OFDM MIMO systems.

## Figures and Tables

**Figure 1 sensors-22-07092-f001:**
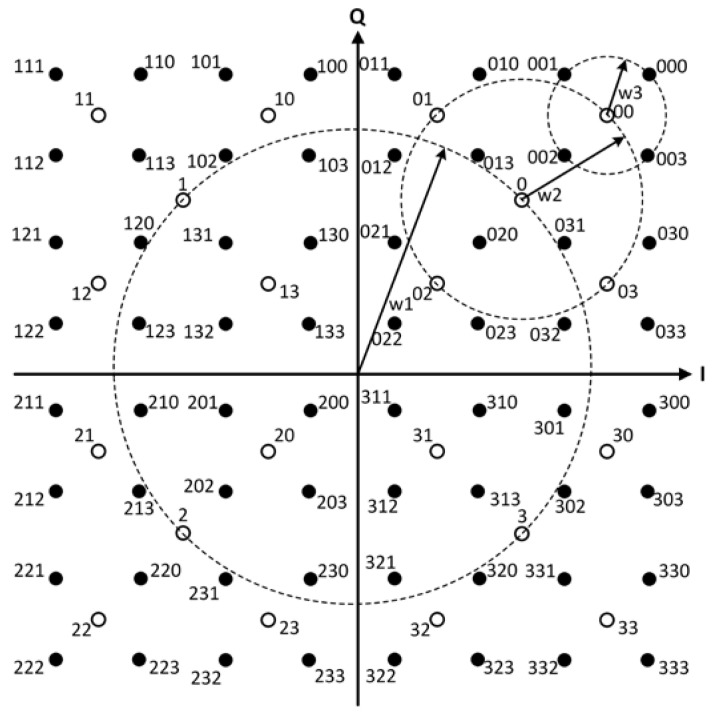
Construction of a 64-QAM constellation with three QPSK constellations.

**Figure 2 sensors-22-07092-f002:**
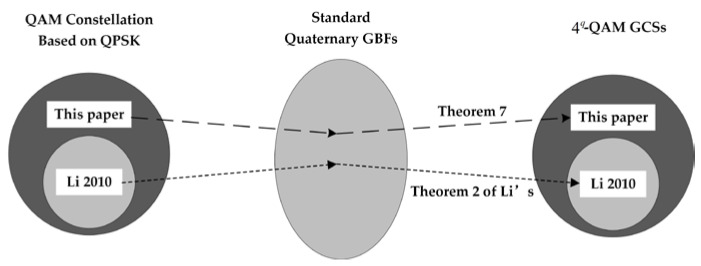
The process from the QAM constellation to the QAM GCSs, and the relationship between [23] and this paper.

**Figure 3 sensors-22-07092-f003:**
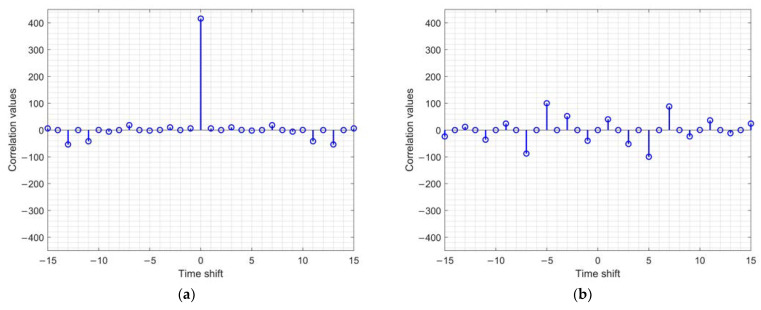
The autocorrelation functions of the proposed 64-QAM GCSs in (39): (**a**) Real part of ***A***’s autocorrelation function $C_{A}\left(\tau \right)$; (**b**) imaginary part of ***A***’s autocorrelation function $C_{A}\left(\tau \right)$; (**c**) real part of ***B***’s autocorrelation function $C_{B}\left(\tau \right)$; (**d**) imaginary part of ***B***’s autocorrelation function $C_{B}\left(\tau \right)$; (**e**) real part of the sum of ***A*** and ***B***’s autocorrelation functions $C_{A}\left(\tau \right)+C_{B}\left(\tau \right)$; (**f**) imaginary part of the sum of ***A*** and ***B***’s autocorrelation functions $C_{A}\left(\tau \right)+C_{B}\left(\tau \right)$.

**Figure 4 sensors-22-07092-f004:**
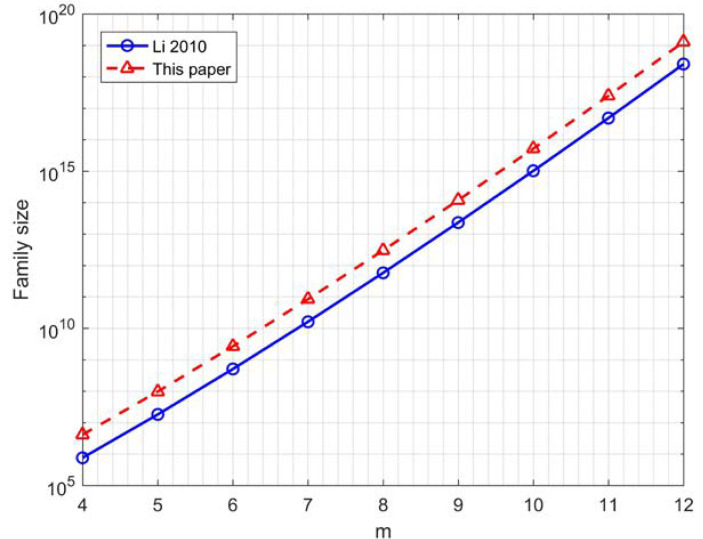
Comparison of the family sizes of 16-QAM GCSs when q=2 between [23] and this paper.

**Figure 5 sensors-22-07092-f005:**
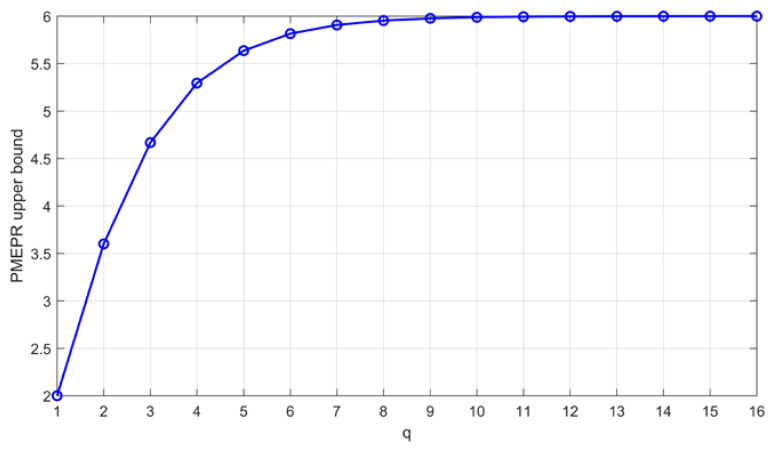
PMEPR upper bound of $4^{q}$-QAM GCSs constructed from Theorem 2, which approaches 6 as *q* increases.

**Table 1 sensors-22-07092-t001:** Offset coefficients that satisfy $2d_{0}^{\left(p\right)}+d_{1}^{\left(p\right)}+d_{2}^{\left(p\right)}=0\;\left(mod\;4\right)$.

NO.	$\left(d_{0}^{\left(p\right)},d_{1}^{\left(p\right)},d_{2}^{\left(p\right)}\right)$	NO.	$\left(d_{0}^{\left(p\right)},d_{1}^{\left(p\right)},d_{2}^{\left(p\right)}\right)$	NO.	$\left(d_{0}^{\left(p\right)},d_{1}^{\left(p\right)},d_{2}^{\left(p\right)}\right)$	NO.	$\left(d_{0}^{\left(p\right)},d_{1}^{\left(p\right)},d_{2}^{\left(p\right)}\right)$
1	(0, 0, 0)	5	(1, 0, 2)	9	(2, 0, 0)	13	(3, 0, 2)
2	(0, 1, 3)	6	(1, 1, 1)	10	(2, 1, 3)	14	(3, 1, 1)
3	(0, 2, 2)	7	(1, 2, 0)	11	(2, 2, 2)	15	(3, 2, 0)
4	(0, 3, 1)	8	(1, 3, 3)	12	(2, 3, 1)	16	(3, 3, 3)

**Table 2 sensors-22-07092-t002:** Comparison of the code rates and family sizes of 16-QAM GCSs when q=2.

Reference	[23]	[28]	This Paper
Family Size	$\left(12m+14\right)\cdot \left(m!/2\right)4^{m+1}$	$\left(58m+105\right)\cdot \left(m!/2\right)4^{m+1}$	$\left(58m+105\right)\cdot \left(m!/2\right)4^{m+1}$
Code Rate	$log_{a}\left|C\right|/2^{m}$	$log_{a}\left|C\right|/2^{m}$	$log_{a}\left|C\right|/2^{m}$
m=2	2.812	3.446	3.446
m=3	1.904	2.213	2.213
m=4	1.221	1.400	1.400
m=5	0.754	0.829	0.829

## Data Availability

The data used to support this study will be available from the corresponding author on reasonable request.

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
