# Peer review of "A New Construction of 4q-QAM Golay Complementary Sequences"

_sensors, 2022, doi:10.3390/s22187092_

Round 1

Reviewer 1 Report

Reviewer Report

Examination of the manuscript shows that the present investigation is not good in the present form. but, the following major points should be addressed carefully during the revision to change my decision in the revision:

-    Add a nomenclature table with SI units and all used abbreviations.

-    Respect the guidelines of the journal and its citation style.

-    Provide a suitable reference for each used equation or model.

-    The main findings should be highlighted in the abstract.

-    The main objectives of the study should be itemized at the end of the introduction.

-    The authors have invited to incorporate real images for the realized experiences.

-    Correlate the main graphical results by an accurate relationship.

-    Improve the discussion more.

-    Link the title with the abstract and conclusions.

-    Remove all typos and grammatical errors.

-    Based on your results, how can the investigators increase the quality of these performance parameters?.

- the paper language should be revised carefully.

Reviewer 2 Report

Dear authors,

The authors presented a new construction of 4?-QAM Golay complementary sequences. This issue is crucial for orthogonal frequency division multiplexing (OFDM) systems. The solution allows to achieve higher data rate and lower peak-to-mean envelope power ratio (PMEPR). The construction obtained through generalized Boolean functions. The upper bound of PMEPR and the family size have been given for this new construct. For one of the comparisons, it was indicated that the new sequences have a larger the family size, which results a large improvement in the data rate without affecting the performance of PMEPR.

Remarks:

 1. In the introduction, the issue of the influence of the PMEPR value on the parameters of the OFDM system has been described in a general way. No reference was made to what PMEPR values are of interest to the authors or where such a system will be used. 

2. Only 7 items out of 37 in References have been published in the last 5 years (2017-2022). Do they fully reflect recent advances on this subject?

Reviewer 3 Report

Recommendations

In this article, “A New Construction of ??-QAM Golay Complementary Sequences. The topic is very important, I think that this contribution does not rise to become a full contribution to your journal, so my advice I recommend improving the future challenges in terms of lower peak-to-mean envelope power ratio in OFDM MIMO system. However, I have some comments and suggestions for the authors. as follows:

1-       The authors must write the full words for abbreviation.

2-       No need to write the equations inside the abstract.

3-       A lot of review studies in terms of this topic  "A New Construction of ??-QAM Golay Complementary Sequences" what's the new contributions compare the related study with the latest published articles in [2021] and [2022].

4-        What’s happens if an increased number of new GCSs.

5-       The authors need to update the literature with some diversified works. Following works can be helpful along with others.

a.     Liu Z, Li Y, Guan YL. New constructions of general QAM Golay complementary sequences. IEEE transactions on information theory. 2013 Aug 15;59(11):7684-92.

6-       The authors must focus on future challenges in lower peak-to-mean envelope power ratio by improving or reducing the computational complexity of OFDM MIMO systems.

b.     Wang Z, Xue E, Gong G. New Constructions of Complementary Sequence Pairs over 4 q-QAM. IEEE Transactions on Information Theory. 2021 Dec 7;68(3):2067-82.

c.     Salh A, Audah L, Abdullah Q, Shah NS, Shipun AH. Energy-efficient low-complexity algorithm in 5G massive MIMO systems. Computers, Materials & Continua. 2021 Jan 1;67(3):3189-214.

7-       The paper lacks a detailed analysis of related works in terms of lower peak-to-mean envelope power ratio by improving or reducing the computational complexity of OFDM.

Round 2

Reviewer 1 Report

The PAPER Needs minor revisions before acceptance.

1.    Figure 1 is not clear and look like image. The author need to revised the Figure.1.

2.    What are the advantages of employed technique?

3.    The lack of physical argumentation is a concern that should be rectified in revised version

4. The paper language should be improved

Reviewer 3 Report

1- The future research must be in the end sentence of the conclusion. 

2-The abbreviations must be put at the end of the section for the introduction.
